# Using High-Density SNP Array to Investigate Genetic Relationships and Structure of Apple Germplasm in Bosnia and Herzegovina

**Almira Konjić** [1],*, **Mirsad Kurtović** [1], **Jasmin Grahić** [1], **Naris Pojskić** [2], **Abdurahim Kalajdžić** [2] **and Fuad Gaši** [1],*

1 Faculty of Agriculture and Food Sciences, University of Sarajevo, Zmaja od Bosne 8, 71000 Sarajevo, Bosnia and Herzegovina; m.kurtovic@ppf.unsa.ba (M.K.); j.grahic@ppf.unsa.ba (J.G.)

2 Laboratory for Molecular Genetics of Natural Resources, Institute for Genetic Engineering and Biotechnology, University of Sarajevo, Kampus, Zmaja od Bosne 8, 71000 Sarajevo, Bosnia and Herzegovina; naris.pojskic@ingeb.unsa.ba (N.P.); abdurahim.kalajdzic@ingeb.unsa.ba (A.K.)

* Correspondence: a.konjic@ppf.unsa.ba (A.K.); f.gasi@ppf.unsa.ba (F.G.)

**Abstract:** Apple accessions, currently maintained within the two main ex situ collections in Bosnia and Herzegovina (B&H), have previously been genotyped using microsatellite markers. The obtained molecular data provided insight into mislabeled accessions and redundancies, as well as the overall genetic structure of the germplasm. The available dataset enabled the creation of a core collection consisting of 52 accessions. The reliability and usefulness of microsatellites has made this low-density marker system a norm in studies on apple germplasm. However, the increased access to medium- and high-density SNP arrays, developed specifically for apples, has opened new avenues of research into apple genetic resources. In this study, 45 apple genotypes consisting of 33 diploid core collection accessions from B&H and 12 international reference cultivars were genotyped using an Axiom® Apple 480 K SNP array in order to examine their genetic relationships, population structure and diversity, as well as to compare the obtained results with those calculated on previously reported SSR profiles. The SNPs displayed a better ability to differentiate apple accessions based on their origin, as well as to cluster them according to their pedigree. Calculating identity by descent revealed 16 pairings with first-degree relationships and uncovered the introgression of 'Delicious' and 'Golden Delicious' into the core collection.

**Keywords:** SNPs; SSR markers; *Malus domestica*; population structure; diversity





## 1. Introduction

Genotyping fruit germplasms is an essential step in the process of conservation and utilization of these genetic resources. Namely, it can help verify the genetic identity of the accessions as well as determine the relationships among the accessions [1]. The use of molecular marker techniques has become increasingly valuable for detecting redundancies in collections and developing effective management strategies for conservation efforts [2]. A review by Nybom and Weising [3] reported that DNA markers managed to reveal a much higher number of mislabeled plant accessions compared to traditional pomological characterization. Furthermore, molecular data obtained through genotyping can be used to evaluate the population structure within an investigated collection, as well as to construct a core collection [4]. A core collection, defined as a subset of accessions that best represents the overall genetic diversity within the collection [5], can then be utilized more effectively for cultivar improvement [6].

Out of the many available DNA molecular markers, Simple Sequence Repeat (SSR) markers, also known as microsatellites, are particularly useful for studying genetic diversity due to their abundance, reproducibility, and polymorphism [3]. These markers have shown

their potential as effective tools for managing *Malus* ex situ germplasm collections [7]. So far, SSR molecular markers have been used for a variety of genetic studies on apple germplasm in Iran [8], Netherlands [9], Spain [10–12], Sweden [13], Italy [4], France [14], Denmark [15], Czech Republic [16], Turkey [17], Russia [18] and Norway [19,20].

Apple accessions, currently maintained within the largest ex situ fruit collection in Bosnia and Herzegovina (B&H), have been genotyped using SSR markers in two consecutive studies [21,22]. The mentioned collection, located in northeastern Bosnia as a part of the fruit tree nursery "Srebrenik", was established in 2000, after a countrywide inventory and collecting mission. The two molecular studies included both traditional apple cultivars and several modern international apple cultivars, which served as reference genotypes. A high level of genetic diversity was detected among the collected apple germplasms, while only one synonym and three homonyms were found. Consequent analyses of the genetic structure revealed a clear differentiation between modern and traditional B&H cultivars, as well as a separation between the traditional cultivars introduced during the Ottoman Empire and the ones introduced later during the Austro–Hungarian rule of the country. An additional ex situ collection ("Goražde") was established in eastern Bosnia in 2014, after a molecular study on apple genetic resources present in that region revealed further traditional apple genotypes not previously collected [23]. Merging the SSR profiles from all apple accessions genotyped in these three studies enabled the detection of 14 different homonyms and 12 synonyms. Furthermore, the molecular data obtained from the studies allowed for the construction of a virtual apple core collection consisting of 22 accessions maintained in the northeastern ("Srebrenik") and 30 accessions conserved within the eastern ("Goražde") ex situ collections.

As shown above, low-density DNA markers such as SSRs have proved very useful in obtaining insight into redundancies, mislabeling, genetic relationships, structure and overall diversity within the analyzed germplasm. However, medium- and high-density marker systems, such as single nucleotide polymorphisms (SNPs), allow for genome-wide comparisons which can reveal small genetic differences that are undetectable with microsatellites. SNPs specifically have several advantages over SSRs including higher resolution, a larger number of markers, easier scoring and wider applicability [1]. The popularity of SNP markers has been increasing due to their flexibility and relative cost efficiency [24].

Earlier efforts in the development of medium- and high-density SNP markers in apples include the Illumina Infinium® II 8 K SNP array [25], as well as the Illumina Infinium® 20 K SNP array developed by Bianco et al. [26]. Despite an improvement in terms of density compared to initial SNP markers developed in apples [27], the use of a 20 K SNP array for screening of a diverse germplasm indicated very rapid linkage disequilibrium (LD) decay for each apple chromosome [1]. A major upgrade, which addressed the issues related to lower marker density, was the development of the Axiom® Apple 480 K SNP array [28]. Namely, the use of a higher density SNP array enables the full extent of genetic variation to be captured across the whole genome [1].

All three previously mentioned apple SNP arrays (8 K, 20 K and 480 K) have so far found a wide range of uses in genetic analyses of apple cultivars, such as: elucidating the pedigree of 'Honeycrisp' cultivar [29], detecting polyploid and aneuploid individuals in an apple germplasm collection [30], tracing founder haplotypes [31], investigating the hybridization between *M. domestica* accessions and its progenitor species [32], reconstructing pedigrees in apple germplasms [33] and identifying genomic regions related with flowering and ripening periods in apples [34]. Additionally, SNP arrays have recently proved useful in examining the relationships between foreign cultivars and domestic germplasms [35]. SNP markers have also recently been used along with SSRs in studies on genetic diversity and population structure within apple germplasm collections in Denmark [36] and Sweden [37]. Larsen et al. [36] found strong concordance between the two marker systems, while Sätra et al. [37] concluded that although SSRs were useful for preliminary screening, SNPs provided more robust information regarding the analyzed accessions. It is

worth noting that both studies relied on medium-density SNP marker sets (15 K and 20 K). Considering the comparative advantages that SNPs possess over microsatellite markers, it would be beneficial to see what additional insight into the B&H apple germplasms a high-density SNP array could provide.

In this study we analyzed 45 apple accessions—33 accessions from the virtual apple core collection conserved in Bosnia and Herzegovina and 12 international reference cultivars. The accessions were genotyped using an Axiom® Apple 480 K SNP array in order to: (a) examine the genetic relationships, population structure and diversity; (b) compare the obtained results with those calculated on previously published SSR profiles.

## 2. Materials and Methods

### 2.1. SNP Data Genotyping

Overall, 33 of the 52 accessions identified as a core collection of the traditional apple germplasm conserved in the two main B&H ex situ collections ("Srebrenik"—northeastern Bosnia and "Goražde"—eastern Bosnia) were included in the study (Table 1). Nineteen remaining core collection accessions have previously been identified as triploids [21–23] and were consequently excluded from analyses. In addition, 12 international reference cultivars from the collection "Srebrenik" were also included.

**Table 1.** A total of 45 apple accessions were used in this research, with assignment of each genotype to a genetic cluster for SNPs (K = 2) and SSRs (K = 3), as well as the physical location of the accession: SR and GO.

|  | Accessions | GCs—SNP | GCs—SSR | Ex Situ Collection |
|---|---|---|---|---|
|  | Traditional accessions |  |  |  |
| 1 | 'Amerikanka' | 2 | 1 | GO |
| 2 | 'Babovaca' | admx | 2 | GO |
| 3 | 'Bascenka' | 1 | admx | GO |
| 4 | 'Bihorka' | 1 | admx | GO |
| 5 | 'Bistrica' | admx | admx | GO |
| 6 | 'Bobovec' | admx | 3 | SR |
| 7 | 'Bukovija' | 1 | 2 | SR |
| 8 | 'Butulija' | 1 | admx | GO |
| 9 | 'Crvenka' | 2 | 3 | GO |
| 10 | 'Djulabija' | 1 | 3 | SR |
| 11 | 'Dobric' | 1 | 2 | SR |
| 12 | 'Dobrinjka' | 2 | admx | GO |
| 13 | 'Habikusa' | 1 | 2 | SR |
| 14 | 'Lederka' | admx | admx | SR |
| 15 | 'Lijepocvjetka' | 2 | 1 | SR |
| 16 | 'Misirka' | 1 | 3 | GO |
| 17 | 'Muslimaca' | 1 | admx | GO |
| 18 | 'Pamuklija' | 1 | admx | SR |
| 19 | 'Paradija' | admx | 2 | SR |
| 20 | 'Petrovaca bijela' | admx | 3 | SR |
| 21 | 'Posavka' | 1 | 3 | GO |
| 22 | 'Prisatka' | admx | admx | GO |
| 23 | 'Ranka' | 1 | 2 | GO |
| 24 | 'Rebraca' | 2 | admx | SR |
| 25 | 'Ruzmarinka' | 1 | 3 | SR |
| 26 | 'Samoniklica' | 1 | 3 | SR |
| 27 | 'Sarija' | admx | admx | SR |
| 28 | 'Senabija' | 1 | 3 | SR |
| 29 | 'Simsirka' | 1 | 2 | GO |
| 30 | 'Srcika' | 2 | admx | GO |
| 31 | 'Srebrenicka' | 1 | 2 | SR |
| 32 | 'Sulija' | 1 | admx | GO |
| 33 | 'Zuja' | 1 | 3 | SR |

**Table 1.** *Cont.*

|  | Accessions | GCs—SNP | GCs—SSR | Ex Situ Collection |
|---|---|---|---|---|
|  | International reference cultivars |  |  |  |
| 1 | 'Braeburn' | 2 | 1 | SR |
| 2 | 'Elstar' | 2 | 1 | SR |
| 3 | 'Fuji Nagafu' | 2 | 1 | SR |
| 4 | 'Gala Galaxy' | 2 | 1 | SR |
| 5 | 'Granny Smith' | 2 | admx | SR |
| 6 | 'Melrose' | 2 | 1 | SR |
| 7 | 'Pilot' | 2 | 1 | SR |
| 8 | 'Pink Lady' | 2 | 1 | SR |
| 9 | 'Pinova' | 2 | 1 | SR |
| 10 | 'Piros' | admx | 3 | SR |
| 11 | 'Golden Reinders' | 2 | 1 | SR |
| 12 | 'Topaz' | 2 | 1 | SR |

SR—accessions maintained in the northeastern ("Srebrenik"), GO—accessions maintained in eastern ("Goražde") ex situ collection, located in Bosnia and Herzegovina; admx—admixed accessions.

Fresh leaf samples for SNP analysis were collected from the ex situ collections in 2019 and stored at −80 °C until the analysis. Genomic DNA from the plant tissue was extracted using commercially available NucleoSpin Plant II—a mini kit for DNA from plants (Macherey-Nagel, Dueren, Germany)—following the manufacturer's instructions with subsequent quality control. The samples were consequently genotyped with the Axiom® Apple 480 K SNP array (Thermo Fisher Scientific, Waltham, MA, USA) [28], containing a total of 487.249 SNPs equally distributed across 17 chromosomes of the apple genome.

*2.2. SSR Data*

Microsatellite profiles of the SNP genotyped accessions (both the traditional cultivars comprising the core collection and the international reference cultivars) were obtained from previous studies by Gaši et al. [21–23]. These studies relied on the use of ten microsatellite markers published by Gianfranceschi et al. [38] (CH01H02, CH01H10, CH01H01, CH02C06) and Liebhard et al. [39] (CH04E02, CH05E04, CH02C02a, CH05E03, CH02D08, CH02C02b).

*2.3. Statistical Analyses*

The SNP data for all 45 genotypes were analyzed using PLINK 1.9 software [40]. SNPs were pruned for linkage disequilibrium (LD) using the command "–indep-pairwise 50 5 0.5" with a window size in SNPs of 50, where one SNP was removed from a pair if LD > 0.5, shifting the window by five SNPs and repeating the procedure. Hardy–Weinberg equilibrium and allele frequency were calculated for pruned and non-pruned data to obtain parameters for minor allele frequency (MAF), observed heterozygosity (Ho), expected heterozygosity (He) and inbreeding coefficient (F). PLINK was also used to calculate identity by descent (IBD) using the "–genome" command, as well as Mendelian errors (ME) with the "–mendel-duos" command. The pruned set of 263 K SNPs served as a basis for calculating identity by descent (IBD) and Mendelian errors (ME). In order to identify first-degree relationships, the parameter "PI_HAT" within the software PLINK was used.

Genetic diversity parameters for SSR data including number of alleles (Na), number of effective alleles (Ne), information index (I), observed heterozygosity (Ho), expected heterozygosity (He) and unbiased expected heterozygosity (uHe) were calculated in R studio [41] using "adegenet" [42] and "poppr" [43] packages. Inbreeding coefficient (F) was obtained by using GenAlex 6.503 [44,45]. A two-proportion z-test [46] was conducted to compare genetic diversity parameters for both datasets. In order to detect potential first- (parent/offspring and full-siblings) and second- (half-siblings) degree relationships among the apple genotypes previously identified as having first-degree relationships based on

SNPs, MLrelate software v1 [47] was used on the SSR dataset. The software relies on the maximum likelihood approach.

Euclidean distance matrix for individuals from the pruned SNP dataset was generated in R studio (version 4.2.0) using bitwise.dist function from "poppr" package [43], whereas Euclidean distance matrix for SSR data was computed using PowerMaker software 3.25 [48].

The neighbor-joining (NJ) dendrogram, based on above mentioned genetic distances, was constructed using MEGA 6 software (Molecular Evolutionary Genetics Analysis) [49]. Correlation between the two matrices with datasets was calculated using Mantel test [50], conducted in PAST Software (version 2.16) [51].

Principal component analysis (PCA) was performed on both the SNP and SSR datasets and plotted in R studio [41] using ggplot2 package [52].

Population structure for SNP data was assessed using fastStructure, a Bayesian model-based clustering algorithm for large genotype datasets [53]. The fastStructure software was executed for multiple values of K (ranging from 1–10) using simple prior model to obtain a range of values for the population number. The web-based software "Structure Selector" was used to select and visualize the optimal number of clusters [54].

The population structure for SSR dataset was estimated with STRUCTURE v2.3.4 software [55]. Most probable K value estimation was evaluated using Structure harvester ver. 0.6.1 application [56]. The accessions were assigned to the genetic clusters according to their highest membership coefficient, with the assigning probability (ql) of 80% according to similar studies [12,57,58]. All accessions with the probability of membership below 80% were classified as admixed. The online software "Structure Plot v2.0" [59] was used to generate a bar plot of the results from Bayesian genetic structure analyses.

## 3. Results and Discussion

### 3.1. SNP Polymorphism

After pruning the data for linkage disequilibrium, 263,406 out of 487,163 SNPs remained for further analysis (Table S1). This is a higher number compared to the SNPs remaining after pruning in a similar study on apples by Muranty et al. [33] (253,096) and slightly lower compared to the number of SNPs selected for a GWAS study by Ur-restarazu et al. [34] (275,233) using the same 480 K array. Hardy–Weinberg equilibrium and allele frequency were calculated for pruned and non-pruned data in order to obtain parameters for minor allele frequency (MAF), observed heterozygosity (Ho), expected heterozygosity (He) and inbreeding coefficient (F) (Table 2). The two-proportion z-test between the pruned and non-pruned SNP data on all 45 samples showed no significant differences in minor allele frequency ($z = -0.04$, $p = 0.96426$), observed heterozygosity ($z = -0.19$, $p = 0.84675$), expected heterozygosity ($z = -0.07$, $p = 0.93852$) and inbreeding coefficient ($z = 0.60$, $p = 0.54462$).

**Table 2.** Comparison of sample size (N), minor allele frequency (MAF), observed heterozygosity (Ho), expected heterozygosity (He) and fixation index/inbreeding coefficient (F) for pruned and non-pruned SNP data.

|  | N | MAF | Ho | He | F |
|---|---|---|---|---|---|
| Pruned SNP data | 263,406 | 0.24 | 0.29 | 0.33 | 0.11 |
| Non-pruned SNP data | 487,163 | 0.25 | 0.31 | 0.34 | 0.07 |

A Mantel test with 10,000 permutations was performed to estimate the correlation between the pruned and non-pruned datasets in the form of two Euclidean distance matrices. The analysis showed a strong correlation between the matrices of pruned and non-pruned SNP data ($R = 0.98$, $p = 0.0001$). Considering that no significant differences between the pruned and non-pruned data were discovered, the more manageable dataset pruned for linkage disequilibrium was used for further analyses. Namely, SNPs in strong

LD are typically pruned out of genomic analyses to reduce the burden of "redundant" variants [60].

### 3.2. SSR Polymorphism

The number of alleles per locus ranged from 5 (CH02C02b) to 19 (CH02C02a). The parameters Ho, He and uHe had similar average values and ranged from 0.49 (CH02C02b) to 0.91 (CH02C02a) for Ho, from 0.57 (CH04E02 and CH02C02b) to 0.92 (CH02C02a) for He and 0.58 to 0.93 (CH02C02b and CH04E02, CH02C02a) for uHe. Inbreeding coefficient (F) had the lowest value in locus CH02D08 (−0.04) and highest in locus CH02C02a (0.1) (Table 3).

**Table 3.** Genetic diversity parameters assessed for 10 SSR loci among all 45 analyzed accessions. Sample size (N), number of alleles (Na), number of effective alleles (Ne), information index (I), observed heterozygosity (Ho), expected heterozygosity (He), unbiased expected heterozygosity (uHe) and fixation index/ inbreeding coefficient (F).

| Locus | N | Na | Ne | I | Ho | He | uHe | F |
|---|---|---|---|---|---|---|---|---|
| CH01H01 | 45 | 10 | 7.09 | 2.04 | 0.87 | 0.86 | 0.87 | −0.01 |
| CH05E03 | 45 | 15 | 7.61 | 2.34 | 0.6 | 0.87 | 0.88 | 0.31 |
| CH05E04 | 45 | 12 | 5.39 | 1.99 | 0.73 | 0.81 | 0.82 | 0.1 |
| CH01H02 | 5 | 9 | 3.96 | 1.63 | 0.71 | 0.75 | 0.76 | 0.05 |
| CH02C02a | 45 | 19 | 12.02 | 2.67 | 0.91 | 0.92 | 0.93 | 0.01 |
| CH04E02 | 45 | 9 | 2.32 | 1.29 | 0.53 | 0.57 | 0.58 | 0.06 |
| CH01H10 | 45 | 13 | 4.51 | 1.89 | 0.8 | 0.78 | 0.79 | −0.03 |
| CH02D08 | 45 | 11 | 6 | 2.03 | 0.87 | 0.83 | 0.84 | −0.04 |
| CH02C02b | 45 | 5 | 2.32 | 0.99 | 0.49 | 0.57 | 0.58 | 0.14 |
| CH02C06 | 45 | 14 | 9.35 | 2.41 | 0.89 | 0.89 | 0.9 | 0.01 |
| Average | | 11.7 | 6.06 | 1.93 | 0.74 | 0.79 | 0.8 | 0.06 |

### 3.3. Genetic Diversity

The analysis of polymorphism of SSR (Table S2) and SNP datasets showed that both marker systems were rather informative. The overall inbreeding coefficient was higher for SNPs (0.11) compared to SSRs (0.06) (Tables 2 and 3). A comparative study on *Juniperus* species, which also relied on SNP and microsatellite analyses, reported higher inbreeding coefficient values for SSRs [61], while Emanuelli et al. [62] found similar values for both marker systems in a study on grape genetic diversity. A two-proportion z-test conducted on SSR and pruned SNP data for the 45 apple accessions did not show significant differences for the inbreeding coefficient ($z = 0.82$, $p = 0.40920$) between the marker systems, which is in line with the conclusions of the mentioned grape study.

On the other hand, the same two-proportion test conducted for the observed ($z = -4.08$, $p = 0.00002$) and expected heterozygosity ($z = -4.36$, $p = 0.00001$) showed significant differences between the values obtained using SSRs and SNPs. Namely, the average values for Ho and He for SNP data were 0.29 and 0.33, whereas the average values for SSR data were 0.74 and 0.79—more than twice higher (Tables 2 and 3). Emanuelli et al. [62] also reported similar differences in these two parameters between SNP and SSR data. Considering the predisposition of microsatellites for de novo mutations, higher occurrence of heterozygotes is to be expected.

### 3.4. Genetic Relationships
#### 3.4.1. Cluster Analyses

Neighbor-joining (NJ) cluster analysis was used to group all 45 accessions based both on the SSR (Figure 1) and SNP (Figure 2) data. While the cluster analysis based on SSR data clearly showed three main clusters, with one of the clusters forming four subclusters, in the NJ dendrogram constructed from the SNP dataset, individual clusters were more difficult to distinguish. However, the separation of the international reference

cultivars was more pronounced on the SNP dendrogram. The most visible example of this was 'Pink Lady', which clearly clustered with the traditional accessions in the SSR dendrogram. On the other hand, SNP dendrogram positioned 'Pink Lady' with the cultivars 'Golden Reinders' (a mutation of 'Golden Delicious'), 'Gala Galaxy', 'Elstar', 'Pinova' and 'Topaz'. It is important to note that 'Golden Delicious' is a parent of 'Pink Lady', 'Gala Galaxy', 'Elstar' and 'Pinova', as well as an ancestor of 'Topaz'. The cluster analysis based on SSRs positioned 'Topaz' closer to 'Melrose' and 'Fuji Nagafu', which are direct progeny of the cultivar 'Delicious' and not of 'Golden Delicious'. Furthermore, the SSR dendrogram clustered 'Braeburn' with the offspring of 'Golden Delicious', while the SNP dendrogram clustered this cultivar tightly with 'Melrose' and 'Fuji Nagafu'. Muranty et al. [33] recently revealed 'Braeburn' as a direct descendant of 'Delicious', making it a half-sibling of 'Melrose' and 'Fuji Nagafu'. Overall, nine accessions designated as traditional ('Amerikanka', 'Lijepocvjetka', 'Dobrinjka', 'Crvenka', 'Rebraca', 'Srcika', 'Petrovaca bijela', 'Prisatka' and 'Paradija') clustered closely with the international cultivars. Some of them have been described as more recent introductions from North America ('Amerikanka' and 'Lijepocvjetka') and Western and Eastern Europe ('Dobrinjka', 'Crvenka', 'Rebraca', 'Srcika', 'Petrovaca bijela' and 'Paradija') by Kanlić [63]. This point will be explored further through the investigation of first-degree relationships and genetic structure.

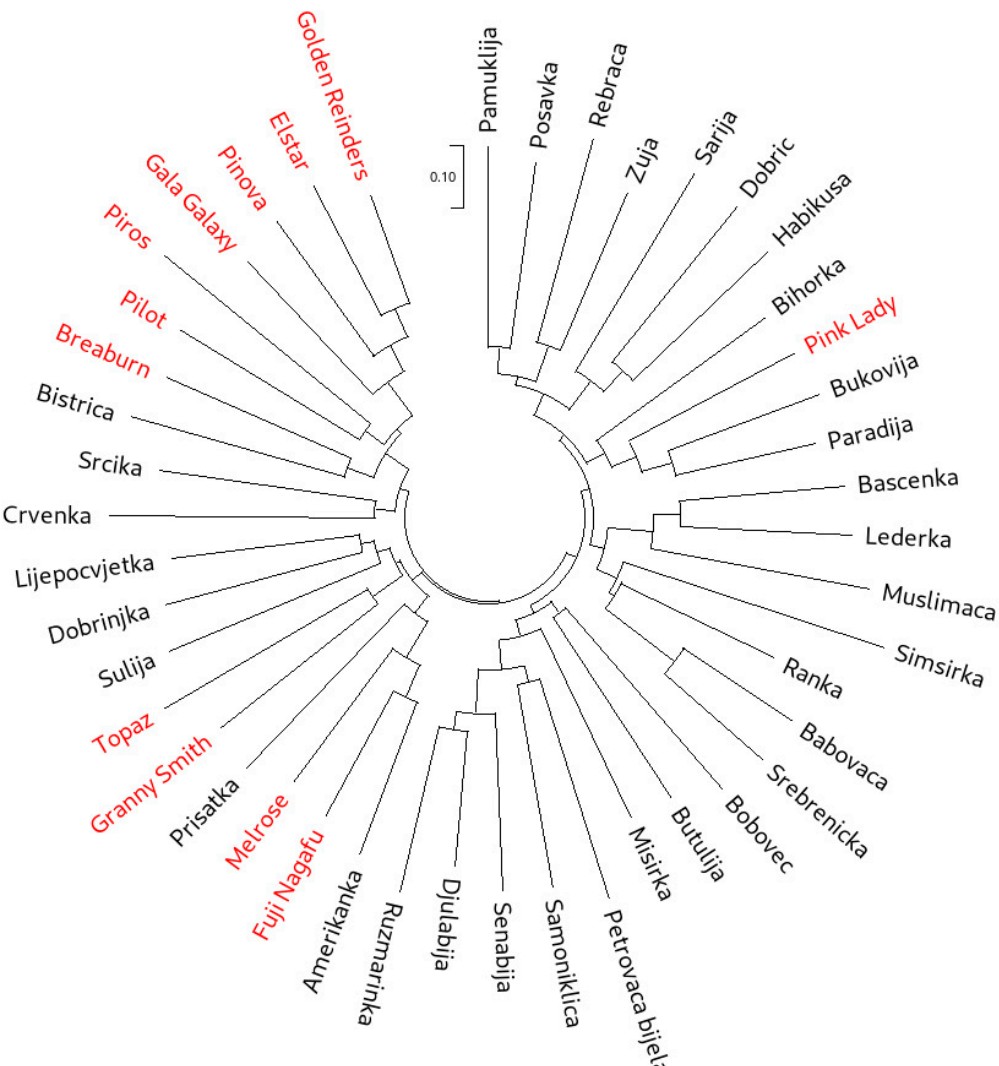

**Figure 1.** Neighbor-joining dendrogram calculated from the SSR dataset on 45 apple accessions (international reference cultivars are marked in red).

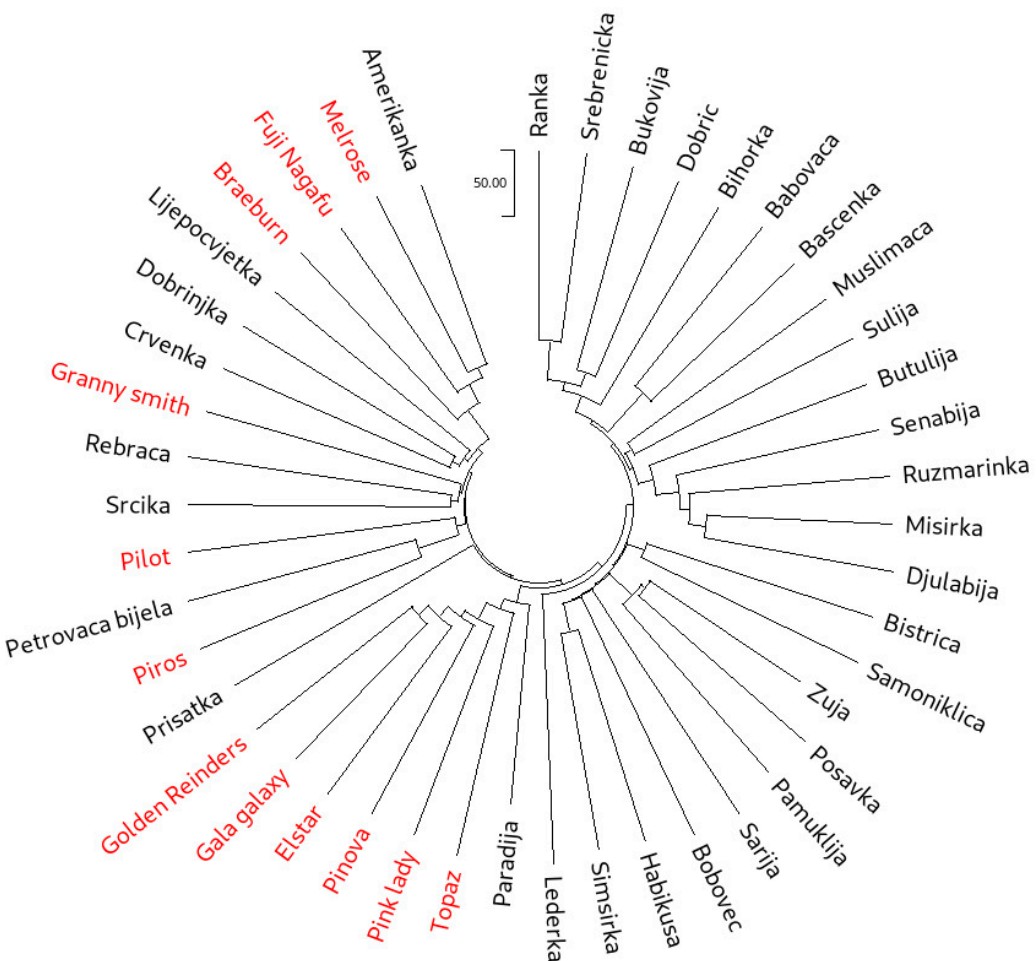

**Figure 2.** Neighbor-joining dendrogram calculated from the SNP dataset on 45 apple accessions (international reference cultivars are marked in red).

The more pedigree-appropriate positioning of 'Pink Lady', 'Topaz' and 'Braeburn' is in line with the fact that SNP markers have been very successful in pedigree network reconstruction. Namely, in the study by Muranty et al. [33] on pedigree relationships among apple genotypes, the use of SNP data helped identify frequent crosses between cultivars from different European regions. SNP data also helped elucidate the pedigree of the 'Honeycrisp' apple through haplotype analysis in a study by Howard et al. [29].

While a high-density SNP array enabled a clear separation between international reference cultivars and a significant portion of traditional B&H apple cultivars, recent use of the same marker system on cultivars from Kazakhstan showed a close relatedness of Kazakhstani germplasm and foreign varieties, with some differentiation through sub-clustering [35]. The differences in the studies should, however, be ascribed to varying levels of introgression of international cultivars into the domestic germplasm. The ability of an SNP array to more clearly differentiate apple accessions based on their origin compared to SSRs has previously been reported by Larsen et al. [36], who attributed this to SNP arrays possessing much higher marker numbers.

A Mantel test conducted in order to check for the correlation between the SNP- and SSR-based distance matrices showed a significant moderate positive correlation between the two (R = 0.4413, *p* = 0.0001). Previous studies on *Citrus* [64], barley [65] and soybean germplasms [66] which combined SNP and SSR marker systems also reported high correlations between the two datasets.

### 3.4.2. Identifying First-Degree Relationships

The pruned set of 263 K SNPs served as a basis for calculating identity by descent (IBD), between all possible pairs among the 45 analyzed accessions. Defining the "PI_HAT" value threshold for first-degree relationships was conducted using international reference accessions with well-known pedigree. Namely, the "PI_HAT" values obtained between 'Golden Reinders' and the direct offspring of its original clone 'Golden Delicious' ranged from 0.368 ('Pinova') to 0.470 ('Gala Galaxy') (Table S3). The defined range was similar to the 0.4 threshold for the "PI_HAT" value set by Muranty et al. [33], who used the same 480 K SNP array.

Twelve pairings were detected in the described range, while four additional pairs of apple genotypes displayed even higher values for this parameter (0.425–0.471). Overall, 21 different apple genotypes were involved in these pairings. Using PLINK, we counted all the Mendelian errors (ME) among all the pairings that cleared the set "PI_HAT" threshold. Again, the threshold for Mendelian errors was set using the ME values obtained for the pairs with well-known pedigrees ('Golden Reinders' and the four direct offspring of its original clone 'Golden Delicious'). All 16 pairs cleared the set ME threshold. Among the mentioned pairs, eight formed between various traditional accessions, four among international reference cultivars and four between the international and traditional B&H apple genotypes. The pair of traditional and international apple accessions that displayed the overall highest "PI_HAT" value was 'Paradija' and 'Golden Reinders'. In fact, the obtained value was even higher than the ones calculated between 'Golden Reinders' and the direct offspring of its original clone 'Golden Delicious'. 'Paradija' also clustered tightly with the 'Golden Delicious' offspring in the SNP dendrogram (Figure 2). The three other pairs of traditional and international apple accessions, with a high "PI_HAT" values were 'Amerikanka' and three different reference cultivars ('Fuji Nagafu', 'Melrose' and 'Braeburn'). As discussed earlier, the apple cultivar 'Delicious' is a joint parent of 'Fuji Nagafu', 'Melrose' and 'Braeburn', making these cultivars half-siblings. This is also in line with the proximity of these genotypes in the SNP dendrogram. Considering that the traditional cultivar 'Amerikanka' displayed "PI_HAT" values higher than the set threshold for first-degree relation with all three half-siblings, it can be concluded that 'Amerikanka' is in fact the cultivar 'Delicious', introduced to Bosnia and Herzegovina probably after World War II.

Regarding the eight pairs of traditional accessions that were according to IBD and ME thresholds in first-degree relationships, the well-known traditional cultivar 'Djulabija' was involved in three such pairings, while the almost equally famous traditional cultivar 'Senabija' formed a trio with 'Zuja' and 'Djulabija'. Considering that the "PI_HAT" value between these two cultivars is very low, it can be suggested that 'Zuja' and 'Djulabija' are the parents of 'Senabija'. Further attempts to reconstruct a pedigree network among the traditional apple cultivars from Bosnia and Herzegovina will require that accessions outside the core collections be genotyped using the same SNP array.

In order to further explore these findings, first- (parent/offspring and full-siblings) and second-(half-siblings) degree relationships, were investigated using the software package MLrelate [47] based on the SSR profiles.

The employed maximum likelihood approach on the SSR dataset revealed twenty-one half-siblings, eight parent–offspring relationships and two full-siblings among the 21 apple genotypes revealed to be involved in first-degree relationships according to the SNP dataset (Table S4). Among the previously described 16 SNP based pairings, seven revealed a parent–offspring relationship based on the microsatellites, with three displaying half-sibling relationships. The six remaining pairs, previously identified as having first-degree relationships based on SNPs, did not show first- or second-degree relationships. Considering the discrepancies between the two datasets (SNPs and SSRs), cultivar pairs with well-known pedigree were investigated.

The cultivars 'Fuji Nagafu' and 'Melrose' were identified as full-siblings based on the SSRs, although they only share one parent in common. Furthermore, 'Melrose' and 'Gala

Galaxy' were determined to be half-siblings, according to the microsatellite data, although these two cultivars do not have a single parent in common. 'Pinova' and 'Elstar', formed a parent–offspring relationship, although they are half-siblings. 'Golden Reinders' and 'Pink Lady' were designated as half-siblings and not parent–offspring. The analyses based on the SSRs did however correctly identify 'Pinova' and 'Gala Galaxy', 'Pinova' and 'Pink Lady', as well as 'Fuji Nagafu' and 'Braeburn' as half-siblings.

Among the traditional accessions, which were identified as having first-degree relationships based on the SNPs, SSR failed to identify first- or second-degree relationships between 'Misirka' and 'Djulabija', 'Djulabija' and 'Ruzmarinka', 'Senabija' and 'Djulabija', and 'Babovaca' and Bascenka', as well as 'Djulabija' and 'Butulija'. Additionally, the SSR data seem to disprove the trio, consisting of 'Senabija', 'Zuja' and 'Djulabija', previously identified with the SNP dataset. In addition, microsatellites failed to identify a parent–offspring relationship between 'Paradija' and 'Golden Reinders' but indicated that 'Paradija' and 'Pink Lady' were half-siblings.

It is important to note that although microsatellite markers represent an excellent tool for identifying relationships between genotypes, high-density markers such as the 480 K SNP array used here possess clear advantages over a ten SSR marker set.

### 3.4.3. Genetic Structure

In order to gain insight into the genetic structure of the analyzed core collection, a Bayesian analysis was implemented on all 45 accessions for both SNP and SSR datasets. The assignment of a genotype to a genetic cluster (GC) was based on a probability of membership, represented by the value ql set above 80%.

ΔK analyses based on the data from 10 analyzed SSR loci, revealed a maximum value for K = 3 (Supplementary Figure S1). Additional peaks, albeit much smaller, were also detected for K = 5 and K = 7. However, they classified almost all accessions as admixed. The first GC included 12 accessions with a ql greater than 80%, while GC2 consisted of 8 accessions and the GC3 comprised 11 accessions. The remaining 14 genotypes were admixed, having probability of membership to any of the three GCs below 80%. Nine of the admixed accessions represented a mixture of GC1 and GC3, while five were a mixture of GC2 and GC3 (Figure 3). GC1 contained all but one of the international reference cultivars ('Piros' clustered in GC3), whereas traditional B&H cultivars grouped dominantly in GC2 and GC3. The only two traditional accessions that grouped in GC1, together with international cultivars were 'Amerikanka' and, 'Lijepocvjetka'. As the name suggests, 'Amerikanka' was assumed to be an old introduction from the north American continent, while 'Lijepocvjetka' is in fact synonym for 'Bellflower' [23]. Previous molecular studies on apple germplasms in Bosnia and Herzegovina by Gaši et al. [21–23] reported that traditional apple accessions closely clustered with international commercial cultivars were likely introduced from the West in more recent past. Meanwhile those traditional accessions that clearly differentiate from international cultivars are thought to have been introduced earlier from the East during the Ottoman Empire. The segregation of many traditional accessions into two distinct groups (GC2 and GC3), could however not be ascribed to the origin and the period of introduction of individual cultivars to B&H. Noticeable mixture of GC1 and GC3 among the admixed accessions indicates a higher level of introgression of international cultivars into this part of the traditional apple germplasm. It is important to note that unlike GC2, GC3 contained one of the international reference cultivars ('Piros'). Additionally, GC3 contained cultivar 'Bobovec' which was previously identified as synonym for 'Bohnapfel' [23].

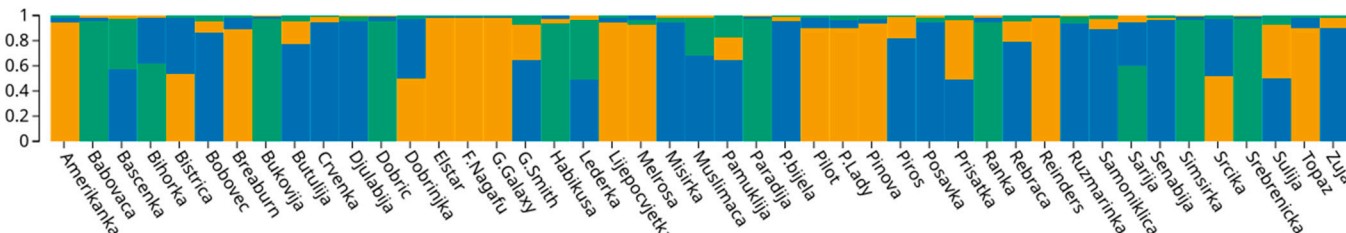

**Figure 3.** Bar plot of the results from Bayesian genetic structure analyses of 45 apple accessions based on SSR loci.

The analysis of pruned SNP data revealed a maximum value for K = 2 (Supplementary Figure S2). GC1 comprised 19 accessions, while the second GC included 17 accessions. The number of admixed genotypes (9) was lower compared to the one obtained in the structure analyses based on the SSR data. Assignment to GCs, based on SNP data showed a clear division between traditional and modern cultivars, without the sub-structuring noted for SSRs. Namely, the Bayesian analysis revealed that GC1 consisted entirely of traditional B&H cultivars, whereas GC2 comprised international cultivars, together with six accessions classified as traditional ('Ljepocvjetka', 'Amerikanka', 'Dobrinjka', 'Rebrača', 'Srčika' and 'Crvenka'). Cultivar 'Dobrinjka' has earlier been designated as a 'Parmenka' ('Pearmain') type of apple originating from the West, while the available sources report that 'Crvenka' originates from Czech Republic [63]. The remaining four accessions have already been described as more recent introductions from the western countries in Section 3.4.1. This is in line with the conclusions by Gaši et al. [23], that distinguished the traditional apple germplasm introduced from Asia Minor and the West. The cultivar 'Paradija', identified as a first degree relative of 'Golden Delicious' (most likely offspring), was admixed, possibly indicating that the genotype is a cross between the international and a traditional cultivar not included in the analyses.

Interestingly enough, fruit from 'Paradija' has been reported to contain some of the highest concentration of polyphenolic compounds among the traditional B&H cultivars, far surpassing international apple cultivars in this regard [67]. Considering Oras et al. [67] reported that traditional apple cultivars are superior to the international ones in polyphenol content, this could serve as an additional argument for 'Paradija' having traditional B&H apple cultivars in its ancestry. All seven mentioned traditional accessions are grouped with the international reference cultivars in the cluster analyses based on SNPs (Figure 2). The second highest K value, based on SNP data, was registered for K = 3. The additional GC only contained 'Piros' and 'Petrovaca bijela', which were admixed for K = 2.

A PCA plot analysis was conducted on both the SNP and SSR datasets (Figure 4), in order to provide a better insight into the differentiation among the accessions assigned to individual GCs. The PC analyses clearly show the benefit of high-density SNP markers in distinguishing between international reference cultivars and the traditional B&H cultivars, with a longer history of cultivation in this country. Larsen et al. [36] also reported that SNPs represented a far more powerful system for detecting structure compared to SSRs, as they managed to detect relatively weak differentiation between Danish and other apple cultivars.

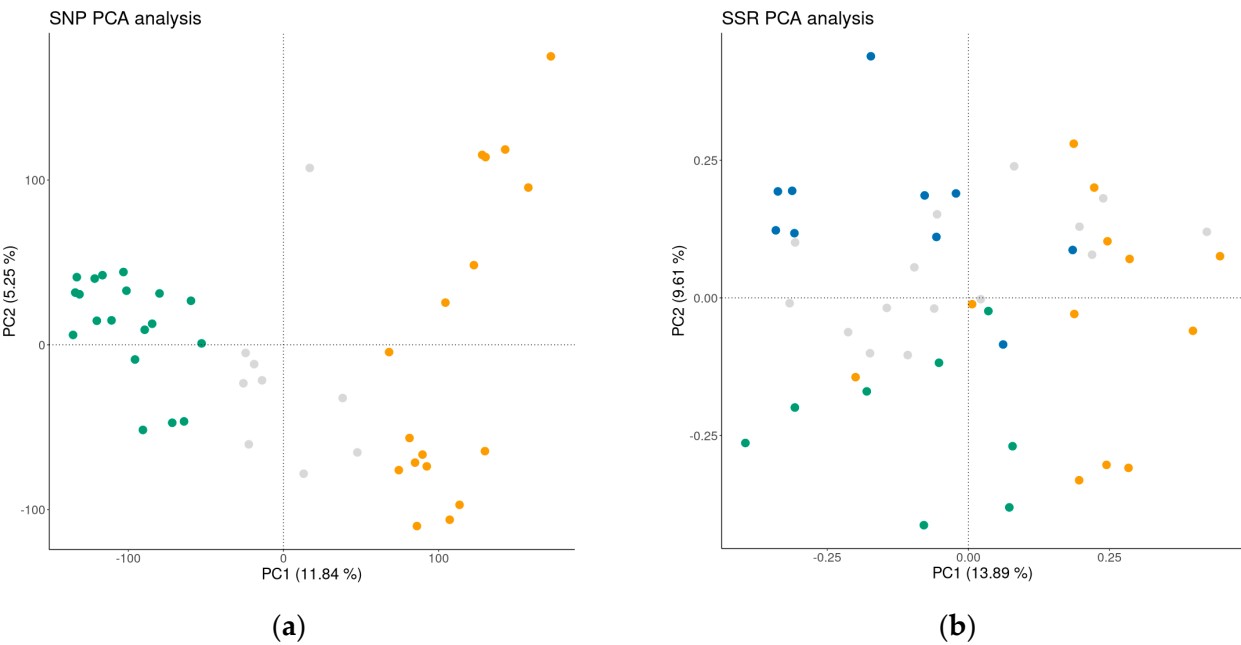

**(a)**                                 **(b)**

**Figure 4.** Principal component analysis of the apple germplasm collection based on SNP loci (**a**) and SSR loci; (**b**) Yellow—international reference cultivars, Gray—admixed, Green (GC2) and Blue (GC3)—traditional apple cultivars.

## 4. Conclusions

To our knowledge, this is the first study to employ a high-density SNP array on fruit germplasm in southeastern Europe. Apple germplasms from this region were notably absent in the Fruitbreedomics project (FP7-KBBE-2010 No. 265582), which developed the Axiom® Apple 480 K SNP array and applied it to apple accessions mostly from western and northern European countries (with the addition of Russia). By publishing the complete SNP genotype data on the diploid core collection from Bosnia and Herzegovina (Table S1), this gap will be reduced.

The array used in genotyping of traditional and international apple accessions displayed clear superiority over the microsatellites, especially in elucidating relationships, but also in differentiating the two germplasms. First, insights into the possible parent–offspring relationships were gained, as well as the confirmation of earlier data on the origin of several apple cultivars designated as traditional. The obtained results paint a picture of a continuous introduction of cultivars into Bosnia and Herzegovina, as well as introgression of international cultivars into the traditional germplasm. In addition, some of the most well-known traditional apple cultivars from B&H, 'Senabija' and 'Djulabija', were revealed to be the most represented accessions in first degree relationships within the core collection. In order to conduct a detailed pedigree reconstruction within the apple germplasm conserved in the two main B&H ex situ collections, accessions outside the core collections need to be genotyped using the same set of SNP markers.

**Supplementary Materials:** The following supporting information can be downloaded at: https://www.mdpi.com/article/10.3390/horticulturae9050527/s1; Table S1: All SNP genotyping data used in the current study; Table S2: All SSR genotyping data used in the current study; Table S3: Material containing obtained PI_HAT values for SNP data; Table S4: Material containing results of parent offspring relationships for SSR data. Figure S1: Plot of delta K values from the Structure analysis based on SSR data on 45 apple accessions and international, reference cultivars. Figure S2: Plot of delta K values from the Structure analysis based on SNP data on 45 apple accessions and international, reference cultivars.

**Author Contributions:** Conceptualization, A.K. (Almira Konjić) and F.G.; data curation, A.K. (Almira Konjić), J.G., A.K. (Abdurahim Kalajdžić) and F.G.; formal analysis, A.K. (Almira Konjić), J.G., N.P.,

A.K. (Abdurahim Kalajdžić) and F.G.; funding acquisition, M.K. and F.G.; investigation, A.K. (Almira Konjić) and F.G.; methodology, A.K. (Almira Konjić), N.P. and F.G.; project administration, F.G.; resources, F.G.; software, A.K. (Almira Konjić); supervision, M.K.; validation, A.K. (Almira Konjić), J.G., N.P. and F.G.; visualization, A.K. (Almira Konjić); writing—original draft, A.K. (Almira Konjić) and F.G.; writing—review and editing, A.K. (Almira Konjić), N.P. and F.G. All authors have read and agreed to the published version of the manuscript.

**Funding:** This research received funding from Ministry of Education, Science and Youth of Sarajevo Canton (Grant No. 11/05-14-27,695-1/19).

**Data Availability Statement:** All SNP data are included in Table S1.

**Acknowledgments:** The authors thank VHLGenetics for providing the SNP genotyping service and especially Jon Wittendorp from the van Haeringen Laboratorium B.V.

**Conflicts of Interest:** The authors declare no conflict of interest.

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
