# Peer review of "Using High-Density SNP Array to Investigate Genetic Relationships and Structure of Apple Germplasm in Bosnia and Herzegovina"

_horticulturae, doi:10.3390/horticulturae9050527_

Round 1
Reviewer 1 Report
1. Line3:it’s better to change “apple germplasm” to “apple core collection”. The author just analyzed 33 diploid core collection apple accessions from B&H. These samples could not enough represent all the apple germplasm in B&H.
2. Line150: delete “minor allele frequency (MAF)”. The author did not perform this kind analysis for SSR data, but add the methods for calculation of the information index(I), Ne, uHe and so on.
3. Line 166-167 and line 169-170: the author should provide the figures of 'Structure Selector' analysis for SNP data and ‘Structure harvester’ analysis for SSR data in order to visualize and prove K=2 and K=3 was the optimal number of clusters for SNP and SSR data, respectively.
4. Line 316-317 and line 351-352:Sentences were incomplete, words were missing.
5. Line 354-395: in this part, the author just only used the SNP data to infer the possible parent-offspring relationships through the parameter ‘PI_HAT’. Actually, the author could also use the SSR data to double check the results obtained from SNP data to make the conclusion more solid. From I know, software such as “Cervus” could easily calculate the possible parent-offspring relationship using Mendel's law of inheritance. So, it is suggested to add parent-offspring analysis based on SSR data.
6. Line 397-435: in this part, the author just described the results of the genetic structure analysis based on SNP and SSR data in words only. For K=2 based on SNP data, it’s easier to understand the ‘admx’. But as to K=3 based on SSR data, the author didn’t make it clear whether the ‘admx’ states were the GC1 and GC2, GC1 and GC3, GC2 and GC3, or a mixture of GC1, GC2 and GC3 together, etc. As far as I know, using the software ‘distruct’ or ‘dapc’ R package can make bar chart to visualize the genetic composition of each sample clearly. It is easier to observe such as “introgression of international cultivars into the domestic germplasm”.
7. Line 489-490: the original SSR genotyping data should also be provided.
8. Line 524, 534, 542, 563, 567, 578, 585, …: the format of the references was not standard, the journal abbreviations should be consistent.
Reviewer 2 Report
Title. The title conveys the main message of the paper — the issues addressed and the relationships among the issues. This answers three important questions: What? Where? How?
Abstract. The abstract is concise, provides a clear overview, includes essential facts for the paper, and concludes with a final point that places the work described in a broader context.
Keywords. These are enough for the topic.
Introduction. The introduction includes background to provide an appreciation for the context of the work presented and also states the rationale and problem that the researchers attempted to answer. In this section, the authors talk about DNA molecular markers (Simple Sequence Repeat, SSR), the genetic characterization of apple, and the purpose of the research.
Material and methods. In this section, the authors describe the correct steps that were followed while conducting their study and explain how they analyzed the data.
Results and Discussion. This section was well written and shows all data with good descriptions. The results say about the objective that motivates the research, and the authors take a broad look at their findings and examine the work in the larger context of the field. However, there are some comments that authors must correct.
For presenting statistics, the conventions are precise and rigid: give the name of the test, give the test statistic value, followed by the degrees of freedom, and the level of significance. I suggest that the authors include all of the information.
Lines 396 to 401 — This paragraph must be move to the material and methods section.
Lines 450 to 452 — This sentence must be move to the material and methods section.
Conclusion. This section included the major conclusions, which were briefly written.
Figures and Tables. Both sections have good information and are necessary for the manuscript, they depict the data nicely.
Figures 1 and 2 — Their descriptions are incomplete.
Reviewer 3 Report
The reviewed paper is devoted to detection of DNA polymorphism in local vs. foreign apple cultivars. This topic is in scope of the journal Horticulturae to which it was submitted. The text is written quite clearly, I have made some corrections and suggestions considering its style and language which can be found in a manuscript file (see attached).
As for the scientific content of this work, I have no major concerns. It is highly recommended to elaborate the Results and Discussion part. At the moment, it is not that easy to understand what new the authors' work brings compared with earlier studies. The originality and novelty of this survey need to be emphasized.
I also suggest to reference some Russian papers on SSR polymorphism of apple trees, such as those conducted by a group of AM Kudryavtsev or this:
Of course, it is not necessary but optional to cover more related papers. However, at the moment the introductory part is fine.
After considering these suggestions and comments, this paper can be recommended for publication.
